# Preclinical Water-Mediated Ultrasound Platform Using Clinical Field of View for Molecular Targeted Contrast-Enhanced Ultrasound

**DOI:** 10.3390/diagnostics15172149

**Published:** 2025-08-26

**Authors:** Stavros Melemenidis, Anna Stephanie Kim, Jenny M. Vo-Phamhi, Edward E. Graves, Ahmed Nagy El Kaffas, Dimitre Hristov

**Affiliations:** 1Department of Radiation Oncology, Stanford University School of Medicine, Stanford, CA 94305, USA; Stavros.melemenidis@cuanschutz.edu (S.M.); anna.kim3@va.gov (A.S.K.); jmv2197@cumc.columbia.edu (J.M.V.-P.); egraves@stanford.edu (E.E.G.); 2Lyons VA Medical Center, 151 Knollcroft Rd, Lyons, NJ 07939, USA; 3Columbia University Vagelos College of Physicians & Surgeons, 630 W 168th Street, New York, NY 10032, USA; 4Department of Radiology, University of California, San Diego, La Jolla, CA 92037, USA

**Keywords:** ultrasound imaging, contrast media, microbubbles, preclinical research, three-dimensional imaging

## Abstract

We report a low-cost protocol and platform for whole-abdomen 3D dynamic contrast-enhanced ultrasound (DCE-US) imaging in mice using a clinical matrix-array transducer. **Background/Objectives**: This platform addresses common limitations of preclinical ultrasound systems. In particular, these systems often lack real-time volumetric and molecular imaging capabilities. **Methods:** Using a modified silicone cup and water bath configuration, mice with dual subcutaneous tumors were imaged in vivo on a clinical EPIQ 7 system equipped with an X6-1 transducer. **Results:** Intravenous administration of targeted microbubbles enabled high-resolution, contrast-mode 3D imaging at multiple time points. Volumetric reconstructions captured both tumors and surrounding anatomy in a single scan, while time–intensity curves and Differential Targeted Enhancement (DTE) analysis revealed greater microbubble uptake in irradiated tumors, consistent with elevated P-selectin expression. **Conclusions:** This standardized imaging platform enables whole-abdomen molecular DCE-US in preclinical studies, facilitating intra-animal comparisons of vascular and molecular features across lesions or organs.

## 1. Introduction

Ultrasound (US) is a non-invasive, cost-effective imaging modality widely used in preclinical research, particularly in mouse models. Mice are the most commonly used species in preclinical research. US is suitable for anatomical and functional imaging of all mouse organs, with the exception of the lungs and head—though both can be addressed with appropriate techniques [1,2,3,4,5]. In contrast-enhanced ultrasound (CEUS), microbubble agents resonate in response to US waves due to their pronounced acoustic impedance mismatch with surrounding tissues. This makes it possible to visualize blood flow and vascular structures in real time. The development of highly echogenic ultrasound contrast microbubbles has drastically expanded the diagnostic potential of US, enabling exclusive imaging of several dynamic tissue perfusion properties and vascular molecular imaging. This enables real-time visualization of tissue perfusion and vascular function, and, when targeted, allows molecular imaging of endothelial markers. With their nonlinear acoustic properties, microbubbles also significantly improve sensitivity and signal-to-noise ratio [6].

Despite these advantages, most preclinical CEUS applications are constrained to 2D imaging, limiting the assessment to a single organ or lesion per acquisition. This is primarily due to the design of high-frequency small-animal transducers, which prioritize spatial resolution at the expense of depth and volumetric capability. Existing workarounds—such as stepwise z-stack imaging or motion-tracking accessories—do not enable real-time 3D acquisition. For example, Fujifilm has a well-developed preclinical 3D-Mode where a distance in Z direction can be set and robotically scan the area, but with small field-size probes and no real-time 3D capability (3D-Mode, FUJIFILM VisualSonics Inc., Toronto, ON, Canada). Another innovation for 3D US imaging is the accessory Piur-Tus-infinity (PIUR tUS Infinity, PIUR Imaging GmbH, Vienna, Austria, used in conjunction with FUJIFILM VisualSonics ultrasound systems) that tracks the probe’s movement when attached to ultrasound transducer and provides tomographic reconstruction, but lacks real-time capability.

Clinical matrix-array probes, although lower in resolution, offer large volumetric fields of view and have shown potential in preclinical contexts when combined with microbubbles. We leveraged such a probe in a water-coupled configuration to enable dynamic, real-time 3D CEUS imaging of the mouse abdomen [7,8,9,10].

Here, we present an in vivo protocol for a platform that enables volumetric imaging of multiple tissues or lesions using a clinical matrix transducer and a single contrast bolus. Our platform configuration uses a water bath to maintain acoustic coupling and thermal support, with the mouse positioned such that its body is submerged while the head remains accessible for anesthesia (dissociative/sedative or general anesthesia with ketamine/xylazine or isoflurane, respectively). In contrast mode, our platform enables simultaneous, dynamic assessment of perfusion across multiple tissues following a single contrast bolus. This methodology captures systemic functional information based on microbubble flow and is compatible with other microbubble formulations. Targeted microbubbles add the potential to assess molecular expression of specific markers on endothelial cells in multiple organs and determine clearance. Capturing both anatomical and functional data, the system can be used for internal control studies with various preclinical diagnostic and pharmacological interests. Our platform aims to showcase that by innovatively integrating existing probe technology and the water-mediated coupling method, we can effectively acquire both contrast and molecular US information, offering comprehensive insights into perfusion dynamics throughout the mouse body. We validated the method using a double tumor mouse model, demonstrating simultaneous assessment of perfusion and molecular marker expression in both tumors. This platform facilitates intra-animal comparisons and expands the capabilities of CEUS for systemic and multi-organ preclinical investigations.

## 2. Materials and Methods

### 2.1. Animal Use and Model

All animal procedures were approved by the Stanford IACUC (APLAC-17186) and conducted in accordance with NIH guidelines. Female Nu/Nu mice (6–8 weeks old; *n* = 10) were inoculated subcutaneously with 5 × 10^5^ LL/2 tumor cells in each upper hind limb. Once tumors reached 120–180 mm^3^, one tumor per mouse was irradiated with a single 6 Gy dose using image-guided SmART system (Precision X-Ray Inc., North Branford, CT, USA).

### 2.2. Tissue Culture and Cell Preparation

LL/2 cells were cultured in DMEM with 10% FBS, passaged twice, and prepared at 5 × 10^6^ cells/mL for injections (100 µL per site).

### 2.3. Imaging Platform

Mice were anesthetized (3% induction, 1.5% maintenance isoflurane, O_2_ at 2 L/min). Tail vein catheterization was performed using a butterfly needle secured with waterproof tape and a trimmed toothpick to seal the entry site. Patency was verified with saline flush (Figure 1A; station A).

A custom imaging station was constructed using a modified silicone cup (76.2 mm tall, 101.6 mm diameter, 12.7 × 15.9 mm window) mounted on a heated tray (36 °C) with a nose cone for anesthesia (Figure 1A; station B). Petroleum jelly sealed the head/limb window. The mouse’s body was submerged in 400 mL of pre-warmed water (37 °C), with a magnetic stir bar placed over the tail to minimize motion (Figure 1B). The magnetic stir bar was used for its shape, weight, and availability, and not for its magnetic properties. A matrix-array transducer (X6-1, Philips Bothell, WA, USA) was suspended vertically into the water for 3D imaging (Figure 1C).

Beyond the ultrasound system itself, which is a capital expense but common to all preclinical imaging workflows, the additional components required to construct the platform are inexpensive and readily available. The custom imaging station consists of a modified silicone cup (<$10), petroleum jelly (<$10 for at least 50 sessions), a heating pad (~$10), and a flexible clamp (<$20). Thus, the overall platform cost is under $50, making it highly affordable and easy to replicate across laboratories.

### 2.4. In Vivo Imaging Protocol

Ten mice were imaged one day before and one day after receiving 6 Gy irradiation to one of two adjacent tumors, resulting in a total of 20 imaging acquisitions.

The US imaging system was configured with an output power of –25 dB and operated at a frame rate of approximately 1 Hz. Time gain compensation (TGC) was aligned in the center, and the dynamic range was set to full (70 dB). Four flash frames were used, with a high pulse repetition frequency (PRF) and persistence turned off. The resolution/general/penetration mode was set to “Res,” with a mechanical index (MI) of 0.09. The focal zone was positioned at 4.0 cm. Flash power was –8 dB, with a flash MI of 0.77. Contrast gain was set to −25 dB.

Tumors were localized in 2D B-mode before 3D contrast-mode imaging (EPIQ 7 system, Philips). Parameters: persistence OFF, flash frames = 3, focal depth = 5 cm. Mice were imaged at discrete intervals: 30 s pre-injection, 2 min post-injection, 30 s at 5 min, and 30 s pre- and 2 min post-destruction pulse. VEVO MicroMarker microbubbles (100 µL; FUJIFILM|VisualSonics, Toronto, ON, Canada) conjugated with anti-CD62P antibodies (ab202983; Abcam, Waltham, MA, USA) were administered IV within 2 h of preparation.

### 2.5. Image Analysis

Volumetric data were analyzed using MevisLab v3.6 (Medical Solutions AG, Bremen, Germany). Time–intensity curves (TICs) were extracted from ROIs, and Differential Targeted Enhancement (DTE) values were calculated to assess targeted microbubble uptake between irradiated and control tumors. DTE is defined as ΔTE = TE_bd_ − TE_ad_, where TE_bd_ is the targeted enhancement before destruction of microbubbles by a high-energy US pulse andTE_ad_ is the targeted enhancement after destruction.

## 3. Results

The protocol was successfully replicated and reproduced across multiple sessions, confirming the feasibility and reliability of the experimental configuration. Mice were maintained under anesthesia for up to 30 min, with a maximum submersion time of 20 min. All animals recovered from anesthesia within 3 min and regained normal activity within 10 min post-procedure. No adverse events or mortality were observed throughout the study.

Volumetric reconstructions of the lower half of each animal were generated from anatomical B-mode ultrasound datasets using MeVisLab, clearly visualizing both adjacent tumors (Figure 2A–H). Tumor volumes were quantified by segmenting serial axial slices (Figure 2A–H; blue and purple regions). Distinct anatomical landmarks, including the left and right kidneys, were readily identifiable.

The imaging field of view (FOV) was configured to include both hindlimb tumors simultaneously, enabling concurrent analysis of contrast dynamics. In studies where tumors are implanted more cranially or in non-tumor-bearing models, the FOV can be adjusted superiorly to encompass the liver and facilitate full abdominal imaging.

Dynamic contrast-enhanced datasets enabled 3D reconstructions for functional analysis, such as tracking contrast agent uptake and clearance over time. Figure 3A–E displays a representative series of coronal slices from a single animal (same as in Figure 2) at multiple time points following intravenous contrast administration. A 30-s baseline was acquired prior to contrast injection (Figure 3A), followed by rapid contrast uptake and enhancement at 2 min post-injection (Figure 3B). Progressive contrast washout was observed over subsequent time points (Figure 3C,D), with signal intensity returning to baseline once microbubbles were cleared (Figure 3E).

Time–intensity curves (TICs) derived from the segmented tumor regions of interest (ROIs) are presented in Figure 4A. The irradiated tumor (right; magenta) exhibited markedly higher contrast enhancement compared to the non-irradiated control, indicative of increased targeted microbubble accumulation. From these TICs, Differential Targeted Enhancement (DTE) values were calculated, demonstrating increase in contrast signal in the irradiated tumor, consistent with elevated P-selectin expression 24 h post-irradiation (6 Gy). Representative fused B-mode and contrast-mode images highlight focal enhancement within both tumors and kidneys, confirming selective binding of the targeted contrast agent (Figure 4(B.0–B.3)).

## 4. Discussion

We describe a low-cost and reproducible protocol and platform for volumetric molecular DCE-US imaging in mice using a clinical matrix-array transducer and a water-mediated configuration. This platform addresses limitations of conventional preclinical ultrasound platforms by enabling real-time 3D acquisition and simultaneous imaging of multiple lesions in vivo. The use of water as a coupling medium minimized acoustic artifacts from air bubbles that often arise with gel-based imaging and are significantly more pronounced in preclinical imaging. Another benefit was increased probe-to-surface distance, which allowed wide-field imaging of two tumors following a single contrast agent bolus. Using a double tumor mouse model, we performed volumetric and dynamic perfusion imaging of both tumors with a single contrast bolus. This also enabled molecular characterization of each tumor. Since both lesions received the same contrast dose, direct intra-animal comparisons were possible. Conventional US systems, limited by small FOV and the lack of true 3D acquisition, cannot provide such multi-organ anatomical and functional data.

Past studies have also used water as a coupling medium, and it has been recently demonstrated in an experimental configuration for ultrafast Doppler ultrasound imaging of the murine kidney [11]. However, in this configuration, a high-resolution probe was used and wide-field, true 3D acquisition was not demonstrated. We aimed to showcase the implementation of low-resolution but sophisticated clinical probes for wide-field 3D DCE-US imaging.

However, as clinical and preclinical probes advance, the capabilities of the concept introduced here can be modified to utilize any 3D imaging probe—matrix or row-column. Clinical probes currently range in frequency from 1–10 MHz and intraoperative probes can achieve 20 MHz; the latter remain limited to spatial resolutions in the hundreds of microns, which can be less effective for certain animal studies. Clinical scanners are typically optimized for human heart rates of 60–100 bpm and lack the temporal resolution required to capture the much faster cardiac cycles seen in preclinical models (400–600 bpm). Preclinical probes can achieve 10–70 MHz with spatial resolutions capable of resolving structures smaller than 100 microns, and preclinical probes are designed with sufficient temporal resolution to resolve cardiac motion within a mouse heart. The Philips XL14–3 achieves even higher frequencies and greater resolutions with a broadband range of 3–14 MHz but does not yet offer contrast mode. Recent results in humans have shown that 3D DCE-US is a powerful and promising imaging method for characterizing and monitoring lesions with reduced sampling errors [12].

Additionally, compared to traditional 2D preclinical systems or robotic stepwise tomographic platforms, our platform offers true volumetric imaging without reconstruction. Although the spatial resolution (~0.5 mm) is lower than high-frequency preclinical probes, the signal quality and contrast-to-noise ratio were sufficient for assessing tumor perfusion and targeted microbubble uptake. This supports applications in multi-organ or multi-lesion studies, particularly when systemic delivery and intra-animal comparisons are essential.

The broader field-of-view and functional imaging capabilities provided by clinical matrix probes—combined with their increasing availability and technological improvements—position them as valuable tools for small animal imaging. Practical configuration considerations for our platform include appropriate sizing of the silicone window, careful sealing for animals of different weights, and elevation of the head to avoid respiratory complications during immersion.

Several limitations should be noted. The spatial resolution achievable with clinical matrix probes (~0.5 mm) is lower than that of high-frequency preclinical probes, which may preclude detailed imaging of small anatomical structures. Additionally, the fixed probe and immersion positioning may not be suitable for imaging targets located outside the abdominal region or in animals that cannot tolerate submersion. Motion artifacts due to respiratory or cardiac cycles may also introduce variability, particularly in studies requiring quantitative perfusion metrics. Lastly, the need for a custom-built imaging station and manual alignment may limit throughput or reproducibility across different laboratory settings.

Future implementations of this platform could explore integration with other imaging modalities, such as photoacoustic imaging, to enhance molecular and structural resolution. Photoacoustic ultrasound captures optical absorption contrast via acoustic detection and may provide complementary data on tumor oxygenation or molecular labeling when combined with contrast-enhanced ultrasound.

As higher-frequency clinical probes become more widely available, the performance of this platform may further improve. Advances in clinical matrix or row-column probes offering higher spatial resolution and broader bandwidth could enable finer structural delineation and greater sensitivity to microbubble signal while maintaining the real-time volumetric capabilities demonstrated in this study.

Despite some tradeoffs in resolution and accessibility, the method presented here offers a scalable and cost-effective alternative to dedicated small-animal platforms, with the added benefit of enabling molecular imaging through clinically available hardware.

## 5. Conclusions

This study presents a low-cost, standardized platform for volumetric molecular DCE-US in mice using clinical matrix-array ultrasound technology. By enabling real-time, wide-field imaging of multiple lesions or organs, the method supports functional and molecular assessment in preclinical models and may stimulate further development and application of volumetric ultrasound technologies in small animal research.

## Figures and Tables

**Figure 1 diagnostics-15-02149-f001:**
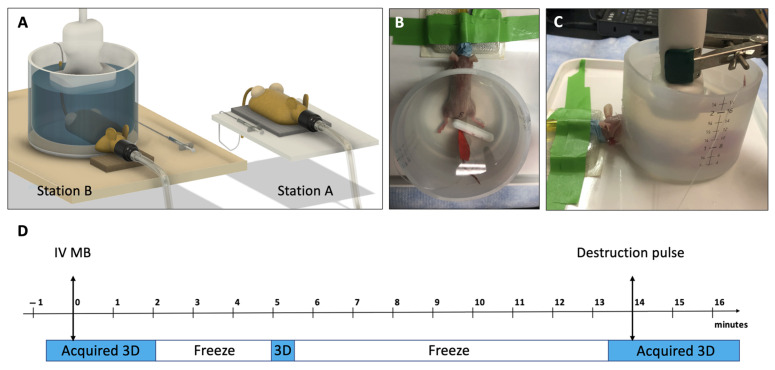
Water-mediated imaging platform and acquisition protocol. (**A**) CAD rendering of the experimental configuration showing Station A (IV catheter placement) and Station B (in vivo ultrasound imaging). (**B**) Top view of the platform with the mouse positioned and submerged in water; red lab tape secures the tail catheter, and a white magnetic stir bar minimizes body motion. (**C**) Side view of the platform showing the ultrasound probe suspended in water via a flexible gooseneck clamp. (**D**) Imaging timeline for a single bolus of targeted microbubbles: 30 s pre-injection, 2 min post-injection, 30 s at 5 min, followed by 30 s before and 2 min after the destruction pulse (total acquisition time: 5 min 30 s).

**Figure 2 diagnostics-15-02149-f002:**
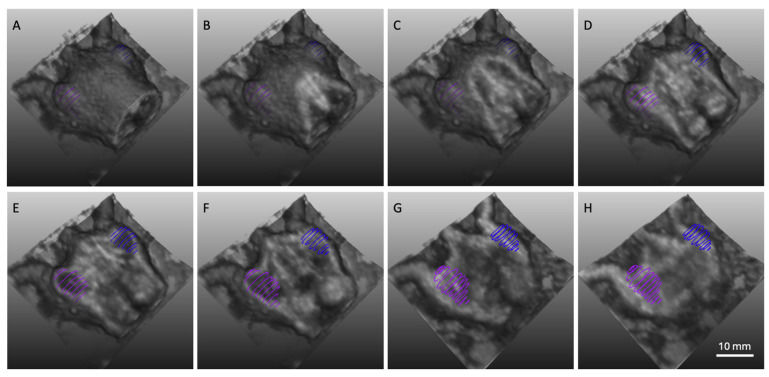
Three-dimensional reconstruction of a B-mode ultrasound dataset from a dual-tumor mouse model, displayed across multiple coronal planes (**A**–**H**). The field of view (FOV) is positioned below the liver to capture both adjacent subcutaneous tumors. Segmentation of the tumors is shown at representative axial planes, with the left tumor highlighted in blue and the right tumor in magenta.

**Figure 3 diagnostics-15-02149-f003:**
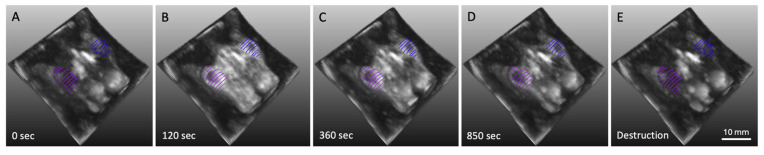
Time-resolved 3D B-mode DCE-US (dynamic contrast-enhanced ultrasound) images of the dual-tumor model, shown at a fixed coronal plane across multiple time points before and after intravenous injection of molecularly targeted microbubbles. (**A**) Baseline image acquired prior to microbubble injection. (**B**) Peak contrast enhancement at 2 min post-injection. (**C**,**D**) Progressive contrast decay over time. (**E**) Return to baseline signal following microbubble destruction.

**Figure 4 diagnostics-15-02149-f004:**
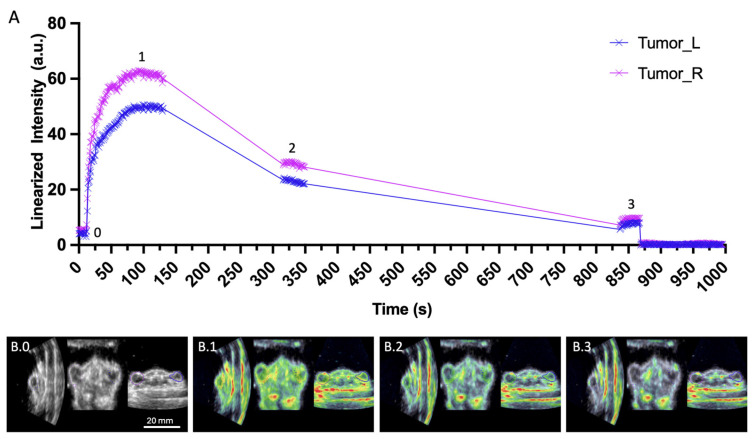
Differential targeted enhancement (DTE) of contrast for adjacent tumors, shown with time–intensity curves and multi-plane B-mode imaging. (**A**) Time–intensity curves (TICs) depicting contrast uptake in both tumors, acquired 24 h after 6 Gy irradiation of the right tumor. Data acquisition was paused between time points 1–3 to conserve storage space. Microbubbles were destroyed after 14 min. (**B.0**–**B.3**) Representative B-mode DCE-US images from three orthogonal planes corresponding to imaging time points 0–3, as indicated in panel (**A**).

## Data Availability

The raw data supporting the conclusions of this article will be made available by the authors on request.

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
