# Peer review of "Preclinical Water-Mediated Ultrasound Platform Using Clinical Field of View for Molecular Targeted Contrast-Enhanced Ultrasound"

_diagnostics, 2025, doi:10.3390/diagnostics15172149_

Round 1

Reviewer 1 Report

Comments and Suggestions for Authors

The article addresses the challenges associated with conducting 3D ultrasound examinations with contrast enhancement in mice. However, it appears that the study requires enhancements to satisfy the rigorous standards for publication. The principal concerns are as follows:

  1. The manuscript contains typographical errors and overall language issues that diminish its clarity. The authors should strive to rectify these to improve the overall readability of the work.
  2. The title suggests that the authors have developed a new ultrasound platform (akin to established research machines like Verasonics), yet the abstract’s opening sentence indicates the introduction of a new protocol. To avoid confusion, the authors should elucidate their use of the term “platform” in the introduction.
  3. The concluding remarks of the Introduction and the initial portion of the Materials and Methods exhibit a considerable redundancy in their descriptions of the imaging platform and its benefits. I recommend a revision that consolidates this information for clarity and coherence.
  4. Certain statements in the Introduction section are inadequately substantiated by pertinent literature sources.
  5. In the Materials and Methods section, the terminology shifts from “platform” to “approach,” then to “method,” and finally to “setup.” This inconsistency creates ambiguity regarding the relationship between these terms. The authors should clarify whether they are presenting a new platform, an approach, or a method.
  6. More comprehensive details regarding the ultrasound imaging system and transducers utilized for both 3D and 2D imaging are required. Key specifications, such as carrier frequency, power settings, gain, number of elements, and matrix dimensions, should be included to enhance the methodological transparency of the study.
  7. In the Results section, the authors refer to “multiple sessions.” It is crucial to specify the exact number of sessions conducted and the total number of mice used in the study to provide a clearer understanding of the experimental design.
  8. The data presented in Figure 4 raises a question: do the curves represent results from a single repetition of the experiment, or were there multiple repetitions conducted? Providing results from several repetitions indicating deviations among the corresponding results would strengthen the findings.
  9. At present, the results are primarily qualitative. To enhance the contribution of this study, the authors should incorporate more quantitative findings and provide comparisons with similar studies in the literature.
  10. The Discussion section fails to address the limitations inherent to the proposed method, which is essential for a balanced and thorough evaluation.
  11. Please, elaborate on how the developed method can be modified or applied to accommodate diverse research scenarios.
  12. In the Reference section, there is a literature source between #9 and #10. Please, check.
  13. Most of the publications in the References section are outdated. Please, consider supporting the relevance of your study by publications that are more modern.

In summary, addressing these concerns will significantly strengthen the manuscript and its potential for publication.

Author Response

The article addresses the challenges associated with conducting 3D ultrasound examinations with contrast enhancement in mice. However, it appears that the study requires enhancements to satisfy the rigorous standards for publication.

The principal concerns are as follows:

We appreciate the reviewers' feedback and the opportunity to improve the clarity and quality of our manuscript.

1. The manuscript contains typographical errors and overall language issues that diminish its clarity. The authors should strive to rectify these to improve the overall readability of the work.

    • We have carefully combed through the manuscript to rectify typographical errors and overall language issues.

2. The title suggests that the authors have developed a new ultrasound platform (akin to established research machines like Verasonics), yet the abstract’s opening sentence indicates the introduction of a new protocol. To avoid confusion, the authors should elucidate their use of the term “platform” in the introduction.

    • We have changed the pertinent sentence in the Introduction to “Here, we present an in vivo protocol that enables volumetric imaging” to avoid confusion. Page 2, line 60.

3. The concluding remarks of the Introduction and the initial portion of the Materials and Methods exhibit a considerable redundancy in their descriptions of the imaging platform and its benefits. I recommend a revision that consolidates this information for clarity and coherence.

    • We agree and have determined that it would be best to go straight into describing the methods rather than recapitulating the overview of the imaging platform and have revised accordingly. (Transition from page 2 to page 3.)

4. Certain statements in the Introduction section are inadequately substantiated by pertinent literature sources.

    • We have revised to ensure that pertinent literature sources are attached adequately to statements in the introduction. (Citations on page 1, line 35; page 2, lines 44 and 59.)

5. In the Materials and Methods section, the terminology shifts from “platform” to “approach,” then to “method,” and finally to “setup.” This inconsistency creates ambiguity regarding the relationship between these terms. The authors should clarify whether they are presenting a new platform, an approach, or a method.

    • We have made extensive changes to address the consistency of the language, particularly using “platform” and “configuration” consistently for different reasons.  (Throughout.)

6. More comprehensive details regarding the ultrasound imaging system and transducers utilized for both 3D and 2D imaging are required. Key specifications, such as carrier frequency, power settings, gain, number of elements, and matrix dimensions, should be included to enhance the methodological transparency of the study.

    • We have added this to the methods section: “The ultrasound imaging system was configured with an output power of –25 dB and operated at a frame rate of approximately 1 Hz. Time gain compensation (TGC) was aligned in the center, and the dynamic range was set to full (70 dB). Four flash frames were used, with a high pulse repetition frequency (PRF) and persistence turned off. The resolution/general/penetration mode was set to “Res,” with a mechanical index (MI) of 0.09. The focal zone was positioned at 40 cm. Flash power was –8 dB, with a flash MI of 0.77. Contrast gain was set to –25 dB.” (Page 4, lines 117-123.)

7. In the Results section, the authors refer to “multiple sessions.” It is crucial to specify the exact number of sessions conducted and the total number of mice used in the study to provide a clearer understanding of the experimental design.

    • We added this to section 2.4: “Ten mice were imaged one day before and one day after receiving 6 Gy irradiation to one of two adjacent tumors, resulting in a total of 20 imaging acquisitions.” (Page 4, lines 115-117.)

8. The data presented in Figure 4 raises a question: do the curves represent results from a single repetition of the experiment, or were there multiple repetitions conducted? Providing results from several repetitions indicating deviations among the corresponding results would strengthen the findings.

    • The curves represent measurements from two tumors implanted in the same animals, each receiving a single bolus microbubble injection at the same time. All animals had tumors on both the left and right hind legs. Using this single bolus injection allowed us to simultaneously compare blood flow in tumors on both sides under identical injection conditions.

9. At present, the results are primarily qualitative. To enhance the contribution of this study, the authors should incorporate more quantitative findings and provide comparisons with similar studies in the literature.

    • Our work does not represent a study but instead aims to demonstrate an imaging protocol that enables imaging of the whole body of an animal, including multiple tumors at the same time following a single bolus injection. Future work will demonstrate how this protocol can be used to carry out complete studies.

10. The Discussion section fails to address the limitations inherent to the proposed method, which is essential for a balanced and thorough evaluation.

    • We agree and have added a paragraph on limitations to the Discussion. (Page 7, line 242-250.)

11. Please, elaborate on how the developed method can be modified or applied to accommodate diverse research scenarios.

    • We have added two paragraphs about this to the Discussion. (Page 7, line 251-260.)

12. In the Reference section, there is a literature source between #9 and #10. Please, check.

    • We have checked and made the appropriate fixes.

13. Most of the publications in the References section are outdated. Please, consider supporting the relevance of your study by publications that are more modern.

    • We have reviewed the references accordingly. While some citations reflect early studies in the field, they were included to acknowledge key developments that underpin current approaches. Our reference list also includes more recent publications that highlight recent advances in 3D DCE-US and its emerging clinical applications. We believe this combination offers a balanced and representative overview of both foundational and contemporary work relevant to our study.

In summary, addressing these concerns will significantly strengthen the manuscript and its potential for publication.

Thank you again for these valuable suggestions. We have made changes in response to all of them.

Reviewer 2 Report

Comments and Suggestions for Authors

General Comments

This manuscript presents a low-cost, reproducible platform for whole-abdomen 3D dynamic contrast-enhanced molecular ultrasound (3D DCE-US) in mice using a clinical matrix transducer and a water-bath setup. The method cleverly addresses the limitations of conventional preclinical ultrasound systems, particularly the lack of real-time 3D imaging and multi-lesion analysis capabilities. The study design is robust, employing a dual-tumor mouse model with one irradiated tumor as an internal control, which convincingly validates the platform's ability to simultaneously assess perfusion and molecular expression (P-selectin). The results are clearly presented, and the conclusions are well-supported. This work is a valuable contribution to the field, facilitating the use of clinical ultrasound equipment in preclinical research.

However, to further enhance the quality of the manuscript, the following revisions are recommended.

Major Comments

  1. Clarification of Methodology:
    • In Section 2.5, "Image Analysis," the authors mention using "Differential Targeted Enhancement (DTE)" to assess targeted microbubble uptake but do not provide the specific formula or method for its calculation. Please add the details of the DTE calculation to allow for accurate understanding and replication of the quantitative analysis.
    • In Section 2.3 (line 106), the use of a "magnetic stir bar" to minimize mouse motion is an interesting and innovative detail. However, its mechanism of action is not entirely clear. Please provide a brief explanation of how it works—for example, is it placed gently on the tail for stabilization, or does it function in another way?

Minor Comments

  1. Wording in Introduction:
    • The phrase "with no additional novelty" in the introduction (line 71), while honest, may diminish the perceived innovation of the paper. It is suggested to rephrase this to emphasize the study's innovation in "cleverly integrating existing clinical technology to solve a specific preclinical imaging challenge." For example: "Our platform aims to showcase that by innovatively integrating existing probe technology and the water-mediated coupling method, we can effectively acquire both contrast and molecular US information..."
  2. Expansion of Discussion:
    • The discussion (Section 4) provides a good summary of the method's advantages (e.g., wide field-of-view, low cost) and limitations (e.g., lower spatial resolution). The paper would be more impactful if it further discussed future applications or potential improvements. For instance, could this platform be combined with other imaging modalities (e.g., photoacoustic imaging)? How might its performance improve with advancements in clinical probe technology (e.g., higher frequency probes)?
  3. Language Polishing:
    • The overall language is fluent, but some sentences could be more concise. For example, the sentence in the abstract, "This approach overcomes common limitations of preclinical ultrasound systems, particularly their lack of real-time volumetric and molecular imaging capabilities," could be split or restructured for easier reading. A final proofread to refine long and complex sentences throughout the manuscript is recommended.

Author Response

General Comments

This manuscript presents a low-cost, reproducible platform for whole-abdomen 3D dynamic contrast-enhanced molecular ultrasound (3D DCE-US) in mice using a clinical matrix transducer and a water-bath setup. The method cleverly addresses the limitations of conventional preclinical ultrasound systems, particularly the lack of real-time 3D imaging and multi-lesion analysis capabilities. The study design is robust, employing a dual-tumor mouse model with one irradiated tumor as an internal control, which convincingly validates the platform's ability to simultaneously assess perfusion and molecular expression (P-selectin). The results are clearly presented, and the conclusions are well-supported. This work is a valuable contribution to the field, facilitating the use of clinical ultrasound equipment in preclinical research.

However, to further enhance the quality of the manuscript, the following revisions are recommended.

We appreciate the reviewers' feedback and the opportunity to improve the clarity and quality of our manuscript.

Major Comments

  1. Clarification of Methodology:
    • In Section 2.5, "Image Analysis," the authors mention using "Differential Targeted Enhancement (DTE)" to assess targeted microbubble uptake but do not provide the specific formula or method for its calculation. Please add the details of the DTE calculation to allow for accurate understanding and replication of the quantitative analysis.
      • We have added the formula to section 2.5: DTE is defined as ΔTE = TEbd − TEad, where TEbd is the targeted enhancement before destruction of microbubbles by a high-energy US pulse andTEad is the targeted enhancement after destruction. (Page 4, line 135-137.)
    • In Section 2.3 (line 106), the use of a "magnetic stir bar" to minimize mouse motion is an interesting and innovative detail. However, its mechanism of action is not entirely clear. Please provide a brief explanation of how it works—for example, is it placed gently on the tail for stabilization, or does it function in another way?
      • The magnetic stir bar was used for its shape, weight, and convenience. No magnetic properties were utilized. We understand that the nature of the description can be misleading. We have added the following clarification to the manuscript: “The magnetic stir bar was used for its shape, weight, and availability, and not for its magnetic properties.” (Page 3, line 100-102.)

Minor Comments

  1. Wording in Introduction:
    • The phrase "with no additional novelty" in the introduction (line 71), while honest, may diminish the perceived innovation of the paper. It is suggested to rephrase this to emphasize the study's innovation in "cleverly integrating existing clinical technology to solve a specific preclinical imaging challenge." For example: "Our platform aims to showcase that by innovatively integrating existing probe technology and the water-mediated coupling method, we can effectively acquire both contrast and molecular US information..."
      • We have made this change. (Page 2, line 72-74.)
  2. Expansion of Discussion:
    • The discussion (Section 4) provides a good summary of the method's advantages (e.g., wide field-of-view, low cost) and limitations (e.g., lower spatial resolution). The paper would be more impactful if it further discussed future applications or potential improvements. For instance, could this platform be combined with other imaging modalities (e.g., photoacoustic imaging)? How might its performance improve with advancements in clinical probe technology (e.g., higher frequency probes)?
      • We have made these additions to the discussion section. (Page 7, line 251-260.)
  3. Language Polishing:
    • The overall language is fluent, but some sentences could be more concise. For example, the sentence in the abstract, "This approach overcomes common limitations of preclinical ultrasound systems, particularly their lack of real-time volumetric and molecular imaging capabilities," could be split or restructured for easier reading. A final proofread to refine long and complex sentences throughout the manuscript is recommended.
      • We have carefully proofread to break up and improve long and complex sentences all throughout the manuscript, including the one you noted. (Page 1, line 14-16.)

Round 2

Reviewer 1 Report

Comments and Suggestions for Authors

The authors provide convincing answers, but several improvements are needed:

  1. Quantitative cost data for the platform should be added to support the cost-effectiveness claim.

  2. Citations are required for the Fujifilm preclinical 3D-Mode and Piur-Imaging information in the introduction.

  3. Scale bars must be included in all figures for accurate dimension assessment.

  4. The meaning of the letters A-H should be described in the caption of Figure 2.

  5. In the line 122 it is stated that the focal zone was positioned at 40 cm. Please, check. 

Author Response

1. Quantitative cost data for the platform should be added to support the cost-effectiveness claim.

  • Silicon cup < $10
  • Petroleum jelly < $10 for at least 50 sessions
  • Heating pad ~ $10
  • Flexible clamp < $20

To support our claim of cost-effectiveness, we added the following: Beyond the ultrasound system itself, which is a capital expense but common to all preclinical imaging workflows, the additional components required to construct the platform are inexpensive and readily available. The custom imaging station consists of a modified silicone cup (< $10), petroleum jelly (< $10 for at least 50 sessions), a heating pad (~ $10), and a flexible clamp (< $20). Thus, the overall platform cost is under $50, making it highly affordable and easy to replicate across laboratories.

Lines 106-111.

2. Citations are required for the Fujifilm preclinical 3D-Mode and Piur-Imaging information in the introduction.

  • 3D-Mode, FUJIFILM VisualSonics Inc., Toronto, ON, Canada
  • PIUR tUS Infinity, PIUR Imaging GmbH, Vienna, Austria, used in conjunction with FUJIFILM VisualSonics (Toronto, ON, Canada) ultrasound systems

These are commercial products.

Lines 52-55.

3. Scale bars must be included in all figures for accurate dimension assessment.

We have added scale bars as necessary.

4. The meaning of the letters A-H should be described in the caption of Figure 2.

We have made the change to the caption of Figure 2.

Line 161.

5. In the line 122 it is stated that the focal zone was positioned at 40 cm. Please, check. 

The focal zone was indeed positioned at 40 cm.

Line 130.